# Parental Self-Efficacy to Promote Children’s Healthy Lifestyles: A Pilot and Feasibility Study

**DOI:** 10.3390/ijerph18094794

**Published:** 2021-04-30

**Authors:** Cayetana Ruiz-Zaldibar, Inmaculada Serrano-Monzó, Olga Lopez-Dicastillo, María Jesús Pumar-Méndez, Andrea Iriarte, Elena Bermejo-Martins, Agurtzane Mujika

**Affiliations:** 1Department of Nursing, Faculty of Health, University of Camilo José Cela, 28692 Madrid, Spain; 2Community Nursing and Midwifery Department, School of Nursing, University of Navarra, 31008 Pamplona, Spain; iserrano@unav.es (I.S.-M.); ebermejo@unav.es (E.B.-M.); 3Department of Health Sciences, Public University of Navarra, 31008 Pamplona, Spain; olga.lopezdicastillo@unavarra.es (O.L.-D.); mj.pumarmendez@unavarra.es (M.J.P.-M.); andrea.iriarte@unavarra.es (A.I.); 4IdiSNA, Navarra Institute for Health Research, 31008 Pamplona, Spain; 5Department of Nursing II, Faculty of Medicine and Nursing, University of the Basque Country (UPV/EHU), 20014 Donostia, Spain; agurtzane.mujika@ehu.eus

**Keywords:** positive parenting, parental self-efficacy, healthy lifestyles, health promotion

## Abstract

Positive parenting programs are a key strategy to promote the development of parental competence. We designed a pilot study based on parental self-efficacy to promote healthy lifestyles in their children aged between 2 to 5 years old. In this pilot study, we aimed to assess the effects of a parenting program on parental self-efficacy and parenting styles. Twenty-five parents were allocated into intervention (N = 15) and control group (N = 10). Parents from the intervention group received four group sessions (120 mi per session) to develop a positive parenting, parenting styles and parenting skills regarding to children’s diet, exercise, and screen time, and two additional sessions about child development and family games. Parents from the control group received these two latter sessions. Parental self-efficacy, parenting styles, and meal-related parenting practices were measured before and after the intervention and at 3-month follow-up. Acceptability and feasibility of the program was also measured. Quantitative data were analyzed using the repeat measures ANOVA and ANCOVA tests and the effect size calculation. Content analysis was used to analyse open questions. Positive trends were found regarding parental self-efficacy and the use of authoritative parenting style. Parents also reported a great acceptability of the program getting high satisfaction. According to the feasibility barriers and facilitators aspects were identified. The positive trends founded in this study support the development of parenting programs to promote healthy lifestyle in children.

## 1. Introduction

Childhood is a period in which speedy physical and psychological development occurs [1]. It is the time for children to attend school, to play and to grow strong and confident with the love and encouragement from their family and an extended community. It is also widely considered to be the right moment to promote the adoption of healthy lifestyles [2,3].

Presently, the social environment we live in, family structures, and way of life are modulating the adoption and maintenance of lifestyles related to diet, exercise, and screen time [4]. This latter issue is of special concern among young people [5]. Physical activity is an important lifestyle aspect throughout childhood as it influences their health and maintenance of a healthy weight [6]. Children are especially sensitive to lack of physical activity due to advances in technology and transportation that have decreased the need for physical exercise in activities of daily living [7]. It is well known that inadequate diet and sedentary lifestyle contribute to health problems in childhood, and therefore pose a public health concern [8,9].

Parents have a significant influence on their children’s healthy lifestyle behaviors through, for example, the provision of healthy food and drink at home, encouraging their children to practice physical activity and by supporting the development of healthy habits or the family meals structure [10,11]. However, the relevance of parents in modulating children’s health behaviors is not limited to that. This vital role comes in the form of providing their children a positive context as well as resources to ensure a healthy development [12]. Several authors have highlighted the importance of developing positive parenting interventions aimed at increasing the competence of preschool parents to promote healthy lifestyles [1,8,9].

Some consolidated studies such as Triple p [13] or Incredible Years [14] showed an impact on aspects related to parenting. These strategies have dealt with resolving parental psychosocial adjustment, behavioral problems and have had a special focus on populations with low socioeconomic status [1]. However, there is a lack of parenting studies with the aim of promoting healthy lifestyles in children through parental competence that have adopted a universal approach [15]. In doing so, there is a need to develop a rigorous theoretical framework [1].

PUEDES Infancy program is a parenting intervention aimed at improving parental competence to promote healthy lifestyles in children. It set out to contribute both empirically and theoretically to the area of positive parenting in children’s lifestyles, and more specifically relating to parental self-efficacy. Parental self-efficacy has important implications for the healthy development of children [16,17]. As Montigny and Lecharité [18] pointed out, terms such as parental competence or parental self-efficacy have been used interchangeably, adding confusion. Parental self-efficacy is defined as parents’ belief in their ability to perform the parenting role successfully [18] and it is a key element in parental competence [19]. Parental self-efficacy plays an important role in raising children, since high levels of self-efficacy have been related to positive behaviors and better life habits that protect children’s health [20]. Parental self-efficacy is a key construct, acting as a mediator between different parenting skills [1,20,21].

According to Bandura’s theory, there are four sources of information for the development of self-efficacy [22,23]: mastery experience that means to become aware of the achievements achieved, vicarious experience related to learn from experience of others, verbal persuasion which means to influence people to believe they have the capabilities to achieve what they seek and psychology states that are people’s perception of their state of anxiety, humor, their emotional or physical states that affect the interpretation of their own experiences and capacities.

The articulation of the Bandura’s four sources of information into activities to achieve parental self-efficacy is the main construct in the theoretical framework of PUEDES program. Along with this, the framework includes parenting styles (authoritarian, authoritative, permissive, and negligent styles), which are the strategies that parents use in raising their children [24,25] and contribute to the developing of positive or negative family’s healthy lifestyles [26,27,28]. Finally, it integrates the REC recommendations of the Committee of European Ministers [29] for positive parenting.

Therefore, we hypothesized that PUEDES, a positive parenting program that promotes parental self-efficacy for the development of healthy lifestyles in 2 to 5 year old children, would provide positive effects on parental self-efficacy, parenting styles, and family habits around.

## 2. Materials and Methods

### 2.1. Participants and Screening

The study took place in Northern Spain. Participants were recruited from community associations, representing a variety of socioeconomic as well as cultural circumstances. Posters, ads, and WhatsApp were used. Recruitment took place from February to May 2017. The inclusion criteria for participants were that they were (1) parents (biological or not) of children between 2 and 5 years old; (2) be older than 18; (3) having signed the informed consent and (4) able to communicate in Spanish.

Figure 1 illustrates the flowchart of participants according to CONSORT diagram [30]. Of the 31 parents who were registered into the program, six of them were excluded, as they did not attend the introductory session. Finally, 25 were eligible for the study (15 allocated to the intervention group and 10 to the control group). Baseline characteristics of the participants are shown in Table 1. There was no difference in the sociodemographic variables between the groups. The mean age of parents was 39.23 (SD 5.76) years in the intervention group and 37.43 (SD 4.83) years in the control group. According to national statistics, in Spain most women have children at 35 years of age. Taking into account that our study chose fathers and mothers of children between 2 and 5 years old, the age of the participants is in line with national standards [31]. In both groups, most of the participants were female (N = 11, 84.6% Intervention group; N = 6, 60% Control group). In a parenting intervention systematic review indicated that female representation was over 90% in all the studies, so our results included more fathers that usual [15]. In both groups, participants were mostly married or lived as a couple. In addition, the participants generally had a high level of education since approximately half had a university degree. Most of the participants from the intervention group were Spanish (N = 7, 53.8%), the rest were from a variety of countries. On the contrary, in the control group there were more participants of foreign origin than national ones, being 40% (n = 4) Spanish, 40% (n = 4) African and 20% (n = 2) South American. According to the census of the territory where the study was carried out, 10.7% of the population has a migrant origin.

### 2.2. Design and Setting

This was a pilot and feasibility study with two treatment groups (intervention and control) and three measurement points (baseline, post-intervention, and follow-up). This study is registered at ClinicalTrials.gov (Identifier: NCT03698110), was carried out between April and June 2017. According to the complex nature of the intervention the Medical Research Council (MRC) methodological framework for complex interventions was used [32,33]. This paper reports the main results of the pilot regarding its efficacy and effect size, feasibility and acceptability that are an essential part before the definitive trial. Based on the MRC and in line with other similar papers on this stage [34,35], conventional sample size calculation did not apply [32,36].

The study took place in two different institutions: in the Association Core, and the University of Navarra. The former is a non-profit organization whose purpose is to welcome and integrate immigrant families at social exclusion risk. Annually it helps about 400 immigrant families. The association is located in a neighborhood whose population density is 25,756 and has a total of 3178 foreign population. The University, located in a different neighborhood than the association, is a private university with approximately 669 employees. In this neighborhood population density is 23,305 inhabitants, of which 2290 are foreign population. The study took place in those two scenarios. Intervention and control participants were present in both, and participants were not mixed between institutions.

Participants were assigned to treatment groups by simple randomization. One researcher from the team who was not involved in the treatment, generated a random sequence using Research Randomizer© software tool available online. Participants were coded in order of attendance to the information session. The sequence generated was applied to the list of participants coded. In one of the centers, randomization was not feasible due to difficulties in assistance to the information session. Analysis of the differences between the groups regarding its sociodemographic data showed that there were no statistically significant differences between them.

### 2.3. Implementing PUEDES Program

The intervention consisted of six 2-h group sessions held once a week. The sessions were led by a Registered Nurse trained in leading small groups and parenting. The sessions were conducted in small groups of 8–10 participants. Each session involved three different phases given the theoretical grounding of the study based on Bandura’s theory of self-efficacy [22,23], positive parenting following the Recommendation Rec of the European Committee of Ministers [29] and parenting styles [24,25]. The program focused on building parental self-efficacy through experiential activities related to child development, lifestyles (diet, exercise, and screen view) and family games. The nurse had a guiding role in each session where parents expressed their experiences in parenting and developed skills to promote healthy lifestyles.

As can be seen in Figure 2 the entire intervention has three different phases. The first phase includes an introductory session where parents received information about child development, specifically in the period comprehending between 2 to 5 years old. The second phase contains four sessions and is divided into two blocks: A and B. Block A includes two sessions on children´s development between 2 to 5 years, while block B is focused on parenting styles and how those affect the adoption of healthy lifestyles with parents from the intervention group.

The four sessions of the PUEDES program had different activities as follows:First session: participants presented themselves. They played with images cards of everyday life situations were viewed to identify the skills they needed to promote healthy lifestyles in their children. They analyzed videos of family behaviors and identified right aspects for child development. The facilitator recommends healthy family games as cooking together or practice a sport.Second session: parents debate about the family games. Then, they watched series sketches and discussed how to deal with complicated parenting situations.Third session: parents watched parts of films and images with scenes representing different parenting styles. During the debate, the nurse guided them toward strategies that they could adopt to promote healthy lifestyles in a positive and authoritative parenting style.Fourth session: parents practiced role-playing and debate about their improvements and skills earned.

At the end, parents summoned to a last session when they played family games. These was based on quizzes and rewards that families, as teams, had to achieve. At the end of the day, children received a gift.

The material used throughout was flexible in response to the adaptability needed for the intervention context. A booklet for parents was designed so that participants were able to follow the sessions. The booklet included activities with photos and links to videos, space for annotations and links to websites of interest. It also includes guidance as to how parents can play family games related to the topics covered in the sessions, such as cooking in family, practicing active games, relaxing or socioemotional games, and practicing parental abilities. The aim was that parents, based on this example, were able to share some quality time with their children through playing with them while implementing the abilities they worked during the sessions. Besides, the booklet also included blank spaces where children could draw regarding the best moments experienced in their family play moments.

### 2.4. Assessments

Both groups were assessed at baseline (T1), at post-intervention (T2), and three months after the program was finished (T3). The outcome measurements included scales, abilities-based tasks, and open questions. The main informant for the evaluation were parents, but researchers notes in the field diary were also used.

#### 2.4.1. Parental Self-Efficacy (TOPSE)

*Tool to Measure Parental Self-efficacy (TOPSE)* [37] has been used in parenting programs with different cultural, social, and educational backgrounds. Its Spanish version includes eight subscales about emotion and affection, play and enjoyment, empathy and understanding, control, discipline and boundaries, external pressures on parenting, self-acceptance and learning and knowledge. It follows the Likert scale of 0 to 10 points, where 0 corresponds to completely disagree and 10 completely agree. Internal reliability coefficients for the subscales ranged from 0.80 to 0.89, and the overall scale reliability was 0.94. [37].

#### 2.4.2. Parenting Style (4Er)

Parenting styles (authoritative, authoritarian, permissive, or negligent) [24,25] were assessed by means of the scale *Escala de Evaluación de Estilos Educativos (4Er)*. This tool includes 20 items on a Likert scale from 1 to 5, the attitude of parents in the traditional dimensions of educational styles: Affection and communication, demands and control (e.g., “In order to not overwhelm my son, I try not to demand much from him”) [38,39]. The scale reliability coefficient of Cronbach’s alpha is 0.73 [40].

#### 2.4.3. Meals in Our Household (MOH)

MOH [41] measures six domains related to families’ mealtimes. We include those related to the structure of family meals, behavioral problems of children during the mealtime, the use of food as reward, parental concern about child diet, and influence of child’s food preferences on what other family members eat. The reliability coefficient of Cronbach’s alpha of the scale is 0.77.

#### 2.4.4. Comprehensive Feeding Practice Questionnaire (CFPQ)

Two subscales of the CFPQ [42] related to children involvement in meal planning and preparation and parental role model regarding diet were used. The original reliability coefficient of Cronbach’s alpha of the subscales ranges from 0.56 to 0.93.

To use these last tools, they were subjected to a translation and piloting process for use in Spanish.

#### 2.4.5. Quality of the Program (Evaluation System of Positive Parenting Programs)

The *Evaluation System of Positive Parenting Programs* evaluates positive parenting interventions [43], according to the general assessment, objectives, materials, assessment, or ethical aspects of the program through 17 indicators based on a Likert scale, from 1 to 6.

#### 2.4.6. Feasibility an Acceptability

Feasibility and acceptability were measured through parental satisfaction. At the end of the program, parents from the intervention group answered a Likert scale ranging from 0 to 10, where 0 was nothing and 10 was completely regarding their opinion about its need and adequacy, and whether they recommend it to friends or family. Parents also answered some open questions about their opinion on the materials, the facilitator of the sessions and any suggestions for improvement or recommendations on the program. Field notes were also used to identify the barriers and facilitators of the implementation process.

### 2.5. Data Analysis

Quantitative data analysis was completed using the SPSS Version 15 (IBM Corp., Armonk, NY, USA). Normality of the outcome data was examined by the Kolmogorov–Smirnov test. Accordingly, Student *t*-test, Mann-Whitney U, chi-square and Fisher’s exact were used as appropriate. To compare the mean difference between groups over the time, repeat measures ANOVA test was used. ANCOVA test was used to compare groups after controlling the basal variability.

Acceptability and feasibility were analyzed through content analysis, which is a technique for interpreting the participant´s answer with a systematic, objective, replicable and valid nature. Following to Myring [44], the responses to the open questions were analyzed identifying those codes of each question (category) that were repeated more frequently in the responses of the participants. The inductive method was used in the analysis process, which favored the generation of codes in order to understand the data in a holistic way.

### 2.6. Ethical Approval

Ethical approval was obtained from the University of Navarra Research Ethics Committee (Code: 2017.025). This study is registered at ClinicalTrials.gov (NCT03698110). Participants were given detailed information about study procedures and written consent was obtained. Confidentiality on participants’ personal data was safeguarded. All personal data of participants were separated from the study data. Only the principal investigator was able to connect these two data sources and only the main researcher was authorized to have contact with the data.

## 3. Results

All data were cleaned and all variables were assessed to determine the assumptions of analyses of variance prior to conducting analyses. Of the 25 parents who entered the study, some dropouts were observed in the post-treatment and in follow-up, especially in the control group as it can be seen in Figure 1. The analyses were based on 22 participants.

A repeated measures ANOVA was performed, with a statistical significance set at *p* < 0.05, to examine the statistical significance of the change between groups (intervention and control) over time (T1, T2 and T3). The analysis of covariance (ANCOVA) with a statistical significance set at *p* < 0.05 was used to compare groups after controlling the basal variability was also used.

### 3.1. Intervention Effects on Parental Self-Efficacy

The effect of the intervention on parental self-efficacy over time showed a moderate effect size (ηp2 = 0.095), although it did not achieve statistical significance (F (2.32) = 1.69; *p* = 0.209). The intervention group experienced an increase in their scores at T2; however, they were not maintained at follow-up (T3). As shown in Table 2, both groups set out at high levels of parental self-efficacy (scale from 0 to 60). After adjusting for baseline variability, there were no significant differences between the groups at T2. However, a large effect size (*d* = 1.846; 95% CI −1.1 to 4.8) was obtained in the difference between the groups at T3 in favor of the control, which could be caused by the decrease of scores of the intervention group, although it did not achieve statistical significance (*p* = 0.197).

The analysis of repeated measures in the parental acceptance dimension indicated a statistical significance interaction over time between groups with a large effect size (F(2.36) = 3.61; *p* = 0.037; ηp2 = 0.167). These results indicated effects of the intervention on parental acceptance; however, as can be seen in Table 2, despite the fact that the parents of the intervention group increased their scores, these were not maintained at T3.

The effect of the intervention on the Learning and Knowledge dimension was statistically significant between groups (F(2.38) = 3.61; *p* = 0.037) with a large effect size (ηp2 = 0.167) over time. As can be seen in Table 2, the levels of learning and knowledge of the intervention group increased to T2. Once the program was completed, the scores of this group decreased at T3. Meanwhile, the control group had a more unstable pattern. Analysis of means after adjustment for baseline variability (T1) suggested large effect sizes for differences between groups at T2 (*d* = 1.817; 95% CI −1.4 to 5) and at T3 (*d* = 2.271; 95% CI −1.1 to 5.6). However, these differences did not achieve statistical significance.

Results did not show any clear statistical impact of the intervention on other dimensions of parental self-efficacy. However, some of them deserve mention. This is the case for perceived pressure by parents in their parenting. The levels in the intervention group had a progressive decrease, whereas in the control group increased their scores, reflecting that they perceived having more pressure throughout the measurements. In the analysis of repeated measures, it was observed that the interaction between the groups had a large effect size (ηp2 = 0.140), nearing statistical significance (F(2.36) = 2.93, *p* = 0.066). In the mean comparison analysis, after adjusting for baseline variability, it was observed that the difference between the groups in T3 had a large effect size (*d* = 3.501), which indicated that parents of the group intervention considered to have less pressures for their environment in parenting than the control group. These results did not reach statistical significance.

Although the analysis after adjusting for baseline variability did not find statistically significant differences, it should be noted that large effect sizes were identified in all dimensions and in the total score. This indicated differential trends between T2 and T3 between groups for parental self-efficacy and all its dimensions.

### 3.2. Intervention Effects on Parenting Style

Parenting styles were measured through their dimensions: Affection and communication, demands and control. In the first two dimensions a very similar pattern was observed. The intervention group increased the levels of both dimensions at T2 and were maintained in the follow-up. On the other hand, the control group in the affection and communication dimension showed an increase between T2 and T3 and in the requirements dimension, a slight increase in T2 and subsequently their levels decreased at the follow-up.

Results showed that in the control dimension, the effect of the intervention was statistically significant over time (F(2.34) = 3.55; *p* = 0.040) with a large effect size (ηp2 = 0.173). In the repeated measures analysis, it was observed that the effect of the intervention on the dimension of affect and communication between the groups had a moderate effect size (ηp2 = 0.134), although it did not achieve statistical significance (F (2.34) = 2.63; *p* = 0.087).

### 3.3. Intervention Effects on Parental Practice

Scores obtained from the practice regarding meal parent report showed no important changes over time in either group. Likewise, analysis of the different dimensions (meals structure, behavior, problem behavior, food reward, concern, influence, involvement, and role model) did not show any important differences between groups over time. However, in the dimension concern, a large effect size was found (F(2.26) = 2.49; *p* = 0.103; ηp2 = 0.161) since, as can be seen in Table 2, the intervention group had a decrease in their scores over time, while the control group varied. In the analysis after adjusting for baseline variability, a large effect size was found in the involvement dimension (*d* = 0.855).

### 3.4. Feasibility and Acceptability

Parents experienced high satisfaction with the PUEDES program. According to the results, it was very likely that they recommended the program to family or friends, (obtained an average of 9 points out of 10). Participants (see Table 3) highlighted equipment, videos and the booklet developed. They also identified the kindergarten as a key factor to participate in the program. Parents also mentioned the role of the facilitator when conducting the program sessions. Finally, some parents identified changes and improvements in their families after completing the program.

Some of the barriers during the implementation process are those regarding the difficulties on the recruitment. This includes a degree of lack of commitment of parents who did not attend the introductory session despite with signed the informed consent. Also, the lack of time that parents have to do activities with their children it seen as a barrier. On the other hand, some activities must be adjusted to the age of children. Also, some technology problems as the use of adequate internet and spaces could be barriers of the process.

On the other side, some of the positive aspects of the process are the marketing and resources used to enroll parents, the good hosting that the associations made to the project, their implication, the flexibility of the program to adapt some activities according to the participants, the active and participation of parents, the enthusiasm of the facilitator and the low cost of the materials.

### 3.5. Program Quality

Quality was assessed along the study using the Evaluation system of positive parenting programs [43]. The PUEDES program showed high quality as it can be seen in Table 4. Aspects that deserve improvement are those regarding the costs, the adaptation of the program to several languages, the use of a website and the lack of relationship with school context.

## 4. Discussion

The aim of this study was to evaluate the effects, feasibility, acceptability and quality of the pilot PUEDES program. The program showed some benefits in parental self-efficacy as suggested by the moderate effect size (ηp2 = 0.095) over time. Furthermore, some dimensions such as emotion and affection, empathy and understanding, discipline and boundary setting, pressures, self-acceptance and learning and knowledge appeared to suggest some effects. However, those effects were not maintained at the follow-up measures. The same trend was found regarding parenting styles. Besides, measurements regarding the parental practices related to diet did not show a significant improvement. These findings are consistent with previous studies which evaluated similar topics of parental self-efficacy after implementation of a parenting program [45,46,47,48,49].

The baseline scores of total parental self-efficacy were high, which suggested that parents had a high degree of self-efficacy at the beginning of the study. In fact, this data showed higher values compared with other research [50]. These results may have been increased due to social desirability, the tendency to give higher scores in relation to socially adequate responses in questions of parenthood [51,52]. In this line, the lack of parental concern has been identified as another factor that could influence the high parental self-efficacy scores [53]. Some aspects such as culture, socioeconomic status or marital status can influence parental self-efficacy [21]. Parents with few resources may have lower parental self-efficacy [54]. These aspects should be analyzed in greater depth in future studies.

Regarding the differential tendency observed in the groups, both the exposure to the intervention and the tools themselves have been able to contribute to a greater awareness of the parents about their own role [55]. The intervention group worked on their parenting during sessions, so that judgments about their role may have been more adjusted to the reality after the intervention as it occurred in other studies [56].

The lack of continuity of the scores achieved in the follow-up could be suggesting the need for additional activities that favor the maintenance of the levels reached. These results are shared by other studies such as the meta-analysis by Barlow, Smailagic, Huband, Roloff, & Bennett [57]. On the other hand, health promotion interventions in childhood and parenting are not immediate, but their benefits are seen in the long term [9]. It is possible that three months of follow-up are not enough to achieve changes, so the measurement of the follow-up should consider more long-term in future studies.

According to the dimensions that measure parenting styles, the intervention group experienced an increase in their levels of affection and communication in the post-intervention measure. Differences were found in the interaction over time for the control dimension. These results could be related to the sessions in which the development of tasks adapted to children’s age was fostered and control and discipline skills were worked on. Negotiation was one of the important points that favored the adjustment toward an authoritative parenting style [58,59]. The family household could have an impact on the lack of statistical significance in parenting styles since all of the participants were in two family household. This is an important issue because children in two-parent homes are influenced by the combined practices of both parents, but parenting styles can be different between mother and father [60].

According to other authors [61], the importance of developing this type of programs in parents of children at an early age to promote attitudes must be taken into account. In addition, it is considered relevant in the development of the authoritative parental style, the adaptation of the attitudes and demands of parents to the age of their children to promote their development [62]. In this sense, the program has been able to bring some improvements, making attitudes toward their children more communicative and reasoned, where they were able to share responsibilities with their children allowing them to solve their daily problems, favoring the development of their learning, autonomy and personal initiative according to their level of development [63]. Although the results in family communication are promising, one aspect that should be improved in future studies is the evaluation of the quality of the hours that parents spend with their children and its association with parenting styles. Some studies affirm that parents who are more involved in caring and who spend more quality time with their children tend to use an authoritative style [64].

There were certain domains of parenting practice in which the changes were close to significance with a moderate effect size as is the case of those related to problematic behaviors of children, the use of food as a reward or the level of parental concern about behaviors and habits dietetics of their children. These results gave clues that the activities carried out during the sessions in relation to the parental practices that promote adequate eating habits such as watching videos to identify positive and negative strategies, or games where parents expressed their experiences, could have an influence.

The lack of statistical difference in the results could be related to the small sample size. However, the nature of the study involves investigating the preliminary effects of the intervention in order to know the underlying mechanisms according to the methodological framework [33]. In this sense, the effect size identified permits identifying tendencies that inform about important clinical differences, which should be confirmed in future studies with a higher statistical power [65].

Parental satisfaction with the program was very high. Both in their general satisfaction and in their perception of the need and its recommendation, the average values were very high. These data are in line with recent studies such as those by Ramos et al. [66] in a very similar context. Some aspects of the acceptability regarding the parental satisfaction as the perceived impact are very enriching to achieve a tighter evaluation [67,68]. In that sense, the good ratings regarding their opinion about the nurse and the characteristics of the program could have an impact on the attendance and participation in the sessions. In our study, almost 80% of the participants attended all the sessions contrasting with studies that affirm the difficulty in maintaining face-to-face throughout the program [69]. Our positive results may be related to the design of the program since the strategy followed was to provide support to parents through their experiences. A study of what parents expect from intervention programs identified that parents wanted strategies and support to deal with the frustrations of raising their children in healthy habits [70].

The study has certain limitations. Although the participant data were analyzed and compared, the data generalization could be limited by possible selection bias. On the other hand, although the study was carried out some time ago, the results are still valid today as positive parenting programs are currently on the rise. Governments and municipalities increasingly focus on parenting in public provision and policy and schools have increased their provision and also their work with parents [71].

Our study has several strengths. First, this feasibility study demonstrated that a parenting program that promote parental competence in healthy lifestyles, provides promising outcomes for improved parental self-efficacy and parenting styles. The identification of potential barriers and facilitators are key to the implementation in a full-scale study [72]. Second, the MRC is a very useful framework to deal with feasibility studies, when physical, psychological, and environmental aspects, education activities are combined [73,74,75]. Finally, the PUEDES program has a strong theoretical foundation based on literature and theories that are key in the development of this kind of programs as it is in line with other authors [1].

## 5. Conclusions

The PUEDES program showed some benefits in aspects of parental self-efficacy and parenting styles. Furthermore, some dimensions such as emotion and affection, empathy and understanding, discipline and boundary setting, pressures, self-acceptance and learning and knowledge appeared to show clues to some effects. However, those effects were not maintained at the follow-up. The same trend was been found regarding parenting styles. Besides, measurements regarding the parental practices related to the diet did not show a significant improvement.

Regarding acceptability and feasibility of the program, parents expressed high satisfaction with the program and they highlighted the materials, resources, and the role of the facilitator. According to the feasibility, barriers and facilitators were identified throughout the development of the study.

The findings from this pilot and feasibility study found some support for the mechanisms predicted to act as mediating factors in parental competencies that promote healthy lifestyles in children. These findings need to be replicated in other studies with bigger population. Also it is important to continue with the implementation of this study in different settings in order to address the universality of the intervention. Following the structure of the MRC for complex interventions, it is necessary to find out parallel results in a full-scale final randomized controlled trial.

## Figures and Tables

**Figure 1 ijerph-18-04794-f001:**
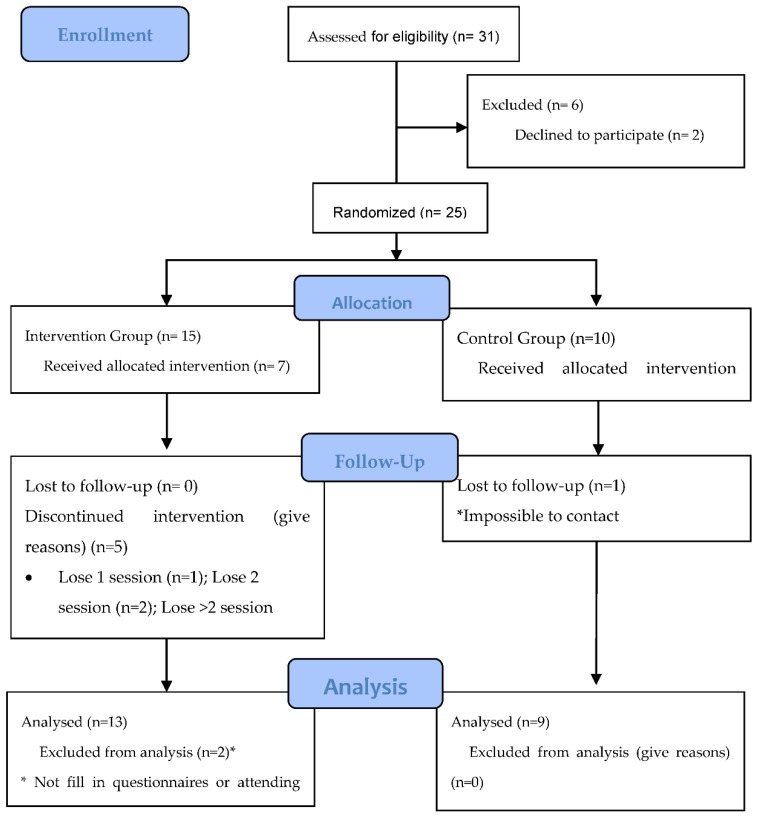
Study participation flow. * Reason for losses.

**Figure 2 ijerph-18-04794-f002:**
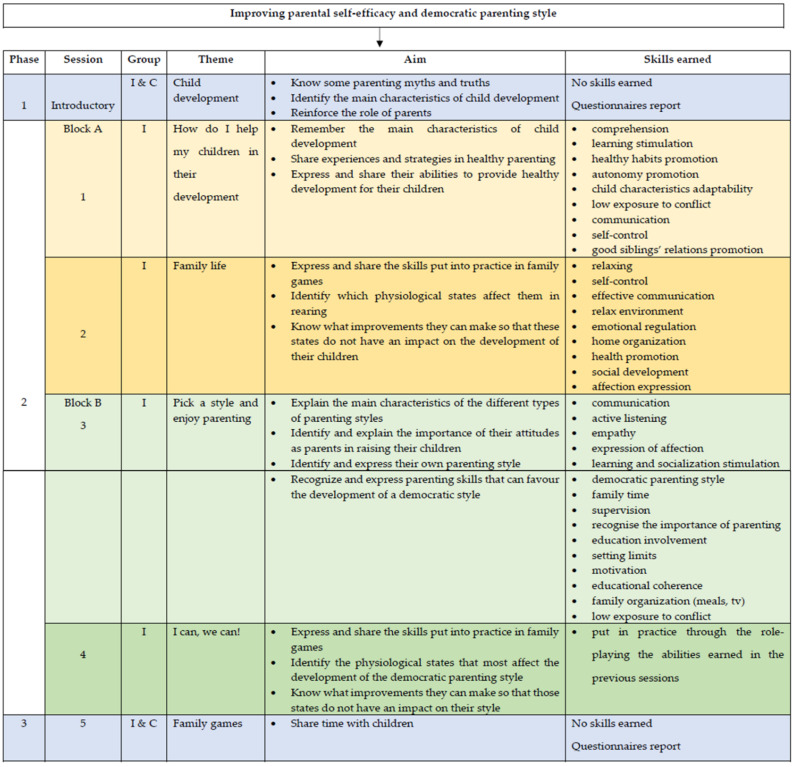
Description of the intervention structure, sessions and skills earned by participants.

**Table 1 ijerph-18-04794-t001:** Participants socio-demographic characteristics.

Variables	Intervention Group (n = 13)	Control Group (n = 10)	*p* Value	95% CI
Age, Mean years (SD)	39.23 (5.76)	37.43 (4.83)	0.491 ^a^	(−3.6 to 7.2)
Female, n (%)	11 (84.6%)	6 (60%)	0.341 ^c^	
Marital Status, n (%)			0.156 ^c^	
Single	0	0		
Married	11 (84.6%)	8 (80%)		
Separated	2 (15.4%)	0		
Divorced	0	0		
Live as a couple	0	2 (20%)		
Studies, n (%)			0.473 ^c^	
Elementary	1 (7.7%)	4 (40%)		
High School	2 (15.4%)	1 (10%)		
Professional Studies	2 (15.4%)	0		
University	3 (23.1%)	3 (30%)		
Master/PhD	5 (38.5%)	2 (20%)		
Country, n (%)			0.309 ^c^	
Spain	7 (53.8%)	4 (40%)		
Other	6 (46.2%)	6 (60%)		
Work activity, n (%)			0.402 ^c^	
Employed	9 (69.2%)	4 (40%)		
Retired	3 (23.1%)	5 (50%)		
Other	1 (7.7%)	1 (10%)		
N° hours of work weekly			0.106 ^c^	
<20 h	4 (44.4%)	0		
20–40 h	5 (55.6%)	3 (75%)		
>40 h	0	1 (25%)		
Annual family income (31.384 euros), n (%)			0.530 ^c^	
Lower	5 (38.5%)	4 (50%)		
Similar	4 (30.8%)	2 (25%)		
Higher	4 (30.8%)	2 (25%)		
People living at home, median	4 (1.5–3)	4 (4–5)	0.372 ^c^	(−1.28 to 1.48)
Children, median	2 (1.5–3)	2 (2–3)	0.326 ^c^	(−1.18 to 1.46)
Experience with children aged 2 to 5 years old, n (%)			0.222 ^c^	
First experience	9 (69.2%)	4 (40%)		
Second experience	4 (30.8%)	6 (60%)		
Time expended with their children during a week, median hours	6 (3.5–7.5)	5 (4.5–12.5)	0.634 ^b^	(−12 to 3)
Time expended with their children during the weekend, median hours	12 (12–21.5)	12 (7–24)	0.920 ^b^	(−12 to 8)

^a^ = student *t*-test with Levene homogeneity of variance test; ^b^ = Mann-Whitney test; ^c^ = Fisher exact test.

**Table 2 ijerph-18-04794-t002:** Changes in TOPSE, 4Er, MOH and CFPQ scores from pre-intervention, post-intervention, and follow-up.

Parental Self-Efficacy	Pre-Intervention (T1) Mean (SD)				Post-Intervention (T2) Mean (SD)				Follow-Up (T3) Mean (SD)			
	n	I	n	C	n	I	n	C	n	I	n	C
Total TOPSE	12	47.73 (2.6)	9	49.01 (3.9)	11	49.26 (3.6)	8	50.03 (5.2) †	12	47.40 (3.4)	8	51.13 (4.1) ƒ
Emotion and affection	13	53.23 (3.3)	10	51.90 (5.0)	12	55.08 (3.5)	9	55.44 (2.4) †	13	54.54 (3.5)	8	53.38 (6.0) ƒ
Play and enjoyment	13	49.15 (5.3)	9	50.67 (4.9)	13	49.15 (5.3)	9	50.67 (5.0) †	13	50.15 (3.4)	8	50.25 (8.4) ƒ
Empathy and understanding	13	47.54 (5.1)	10	49.20 (2.9)	13	51.23 (6.6)	9	45.33 (7.1) †	13	49.85 (5.8)	8	45.75 (6.4) ƒ
Control	13	42.92 (5.2)	10	41.90 (6.0)	13	43.46 (6.1)	9	45.33 (7.1) †	13	41.23 (7.4)	8	45.75 (6.4) ƒ
Discipline and boundaries	13	44.00 (5.1)	10	46.50 (5.6)	13	46.62 (7.3)	9	48.22 (6.9) ‡	13	44.15 (7.4)	8	51.25 (4.5) ƒ
External pressures on parenting	13	49.08 (7.1)	10	44.70 (12.5)	13	46.92 (9.0)	8	45.88 (10.7) ‡	13	46.08 (8.4)	8	50.75 (8.0) ƒ
Self-acceptance	12	48.42 (3.3)	10	51.10 (6.0)	12	51.17 (3.7) *	9	53.67 (5.4) ‡	12	48.75 (6.9)	8	52.63 (3.5) ƒ
Learning and knowledge	13	50.15 (5.0)	10	50.90 (5.2)	13	52.62 (5.1) *	9	48.89 (7.9) ‡	13	48.23 (6.1)	8	52.25 (5.3) ƒ
4Er												
Affection and communication	13	4.34 (0.5)	9	4.44 (0.8)	12	4.51 (0.4)	9	4.30 (0.5) †	13	4.45 (0.5)	8	4.62 (0.4) §
Demands	13	3.98 (0.4)	9	4.00 (0.6)	13	4.15 (0.5)	9	4.10 (0.5)	13	4.15 (0.6)	8	3.82 (0.8) §
Control	13	3.60 (0.6)	9	3.63 (0.6)	13	4.00 (0.6)	9	3.92 (0.7) ‡	13	3.57 (0.6)	6	3.70 (0.7)
MOH & CFPQ												
Structure meals ^a^	12	2.27 (0.3)	10	2.35 (0.1)	11	2.27 (0.4)	9	2.34 (0.2)	12	2.20 (0.3)	8	2.40 (0.4)
Negative behavior ^b^	13	1.32 (0.6)	9	0.91 (0.6)	13	1.14 (0.8)	8	0.89 (0.7) †	13	1.17 (0.7)	8	0.67 (0.4)
Problem behavior ^c^	13	0.80 (0.4)	8	0.55 (0.8)	11	0.66 (0.5)	6	0.50 (0.8)	12	0.74 (0.5)	9	0.60 (0.6)
Food reward ^a^	13	0.90 (0.9)	10	1.19 (1.0)	13	0.81 (0.7)	9	1.04 (0.9) †	13	0.68 (0.5)	9	0.93 (0.7)
Concern ^d^	10	2.01 (0.9)	8	1.42 (0.3)	9	1.76 (0.9)	8	1.66 (0.6) ‡	12	1.38 (0.3)	8	1.59 (0.5) §
Influence ^d^	13	1.25 (0.7)	10	0.96 (0.9)	13	1.12 (0.8)	9	1.23 (0.7)	12	1.10 (0.7)	9	0.96 (0.4) §
Involve ^e^	13	3.64 (0.8)	10	3.33 (0.7)	13	3.62 (0.8)	9	3.52 (0.7) †	13	3.18 (0.8)	9	3.48 (0.9) ƒ
Role model ^e^	13	4.27 (0.7)	10	3.92 (1.0)	13	4.35 (0.7)	9	4.06 (0.7)	13	4.37 (0.7)	9	3.92 (0.6) §

I = Intervention Group; C = Control Group. TOPSE: Tool to Measure Parental Self-Efficacy. Scores can range from 0 to 60, a high score shows a high level of parental self-efficacy. 4Er: Parenting Style. Scores can range from 0 to 5, a high score shows a high level of parenting styles. MOH: Meals in Our Household; CFPQ: Comprehensive Feeding Practice Questionnaire. ^a^ Scores can range from 0 to 4, a high score shows a high level of structure meals and food reward; ^b^ Scores can range from 0 to 4, a less score shows a less level of negative behavior; ^c^ Scores can range from 0 to 3, a less score shows a less level of behavior problems; ^d^ Scores can range from 0 to 5, a less score shows a less level of concern and influence; ^e^ Scores can range from 0 to 5, a high score shows a high level of involve and role model. * *p* < 0.05 from de ANOVA analysis; † ANOVA Moderate effect size; ‡ ANOVA Large effect size; ƒ ANCOVA Large effect size; § ANCOVA Medium effect size.

**Table 3 ijerph-18-04794-t003:** Content analysis results.

Topic	Example of Parental Answered
Equipment	“*Both the materials and the resources have been very useful for me*” (Participant 1) *“**They have seemed very suitable, entertaining and fun”* (Participant 15)
Booklet	*“**I found it interesting to have this support material, both in the classroom activity and for those at home. I think I can continue using it in the future and it will be a reference in possible family situations.”* (Participant 4)
Videos	*“**The videos are very accurate with each topic that was discussed”* (Participant 7)
Nursery	*“The nursery has been indispensable to be able to attend”* (Participant 9)
Nurse role	“*I really liked it a lot. It has made me reflect on my family and my parenting style. Rethink things that I had never stopped to think.*” (Participant 15) *“She gave us an excellent attention, communication, she guided us a lot and helped us to develop ourselves in each session.”* (Participant 19)
Improvements	*“Perhaps it could be a website to complete the information and a meeting with some periodicity on topics of interest to parents.”* (Participant 18)
Changes	“*I have attended several parent groups, several sessions and I think this program has made me think more than any other.*” (Participant 15)

**Table 4 ijerph-18-04794-t004:** Quality of the program.

INDICATORS	TOTAL	%
Implementation		
Institutional management	13/36	36.1%
Cost	4/24	16.7%
Publicity	31/36	86.1%
Community support	NA	
Total	48/96	50%
Methodology		
Overall rating	30/30	100%
Material	29/30	96.7%
Learning methodology	35/36	97.2%
Mixed format	4/18	22.2%
Evaluation	58/66	87.9%
Professional profile of responsible	16/24	66.7%
Ethics aspects	30/30	100%
Total	202/234	86.3%
Content		
Aims	36/36	100%
Contents	32/36	88.9%
Scientific foundations	22/24	91.7%
Linguistic offer	14/24	58.3%
Adjustment of the COPP *	27/30	90%
Family School Coeducation	6/36	16.7%
Total	137/186	73.7%
Final score	387/516	75%

NA: Not Applicable; * COPP: Optimal Curriculum of Positive Parenting.

## Data Availability

The data presented in this study is available upon request to the corresponding author.

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
