# Peer review of "Parental Self-Efficacy to Promote Children’s Healthy Lifestyles: A Pilot and Feasibility Study"

_ijerph, 2021, doi:10.3390/ijerph18094794_

Round 1
Reviewer 1 Report
The paper deals with interesting and rather important topic. However I have a few essential comments to the manuscript. First the convenience sampling was used what may bring the selection bias and the authors not deal with this issue at al. Second ,the study is presented as the pilot and feasibility study and some conclusions are rather preliminary so there is the question if the results may be generalized and if yes to what population Nord Spain, Spain, Europe... . Third, the PUEDES program need to be described in more details, however in the concentrated way. Fourth, authors mention in the paragraph "The intervention took place in two different institutions: in the Association (blinded 139 for review), and the University of (blinded for review). The former is a non-profit organi- 140 zation whose purpose is to welcome and integrate immigrant families at social exclusion 141 risk. Annually it helps about 400 immigrant families. The association is located in a neigh- 142 bourhood whose population density is 25,756 and has a total of 3,178 foreign population. 143 The University, located in a different neighbourhood than the association, is a private uni- 144 versity with approximately 669 employees. In this neighbourhood population density is 145 23,305 inhabitants, of which 2,290 are foreign population." This paragraph is rather confusing, and if I understand correctly the authors works with families in risk of social exclusion not with the general population or they mix it. This need to be clarified. Fifth, the authors works also with the statements, however the quantitative research setting is just stated in the abstract and not in the methodological part. The work with the qualitative methodology is not sufficient at all and the qualitative part is rather unreported. The authors need to make clear why they submitted the research carried out in 2017 in the year 2021.
Author Response
Responses are attached. Than you for the comments.

Reviewer 2 Report
Introduction:
The content should have emphasized on the importance of physical activity as a healthy habit.
The explanation about the Bandura´s theory should be more extended to clarify the menaning of "self-eficacy".
2.4. Assessments
Reasons why it took place only 3 assessments.
Author Response
Responses are attached. Thank you for the comments.

Reviewer 3 Report
The manuscript presents findings of a pilot experimental study that assessed the effects of parenting intervention in strengthening parental self-efficacy for promoting children's healthy development at the early childhood stages. Because children's early experiences are powerful determinants of later developmental trajectories, the study's purpose is an important one. It is a good way of being proactive rather than reactive in promoting healthy development of children. The findings in many ways confirmed what we know to be important to understand various parenting styles and the impact they have on children. In particular, the understanding that resourced context (e.g. family structure, income, education etc) are important for understanding competence-based parenting. Overall, the write up of the manuscript is good. The findings are interesting, but not surprising, and the conclusions drawn are, for the most part, consistent with the findings. There are few comments that I have:
In reviewing Table 1, I noticed that in terms of family structure/marital status, nearly all the participants are in two family household. That is an important context to understand parenting style. Especially two parent household with limited conflict. That maybe one reason explaining the lack of statistical significance between the intervention and the control groups on the parenting styles. Additionally, when observing Table 1, the number of hours spent with children on weekends and weekdays, the range for the control group was wider than that of the intervention group. Here again, the family structure may be playing a role there, which may contribute to the psychological availability of the parent because it is not just the time spent as in physical availability, but the quality of it is what counts most. Research has shown that emotional availability of the parent at this stage is important to affect the parenting style, which in turn affects communication and bonding with the child. So I wonder how some of these nuances may be at play in the observed findings. If they did this study with primarily low resourced as well as single parent household, how might the findings be?
The qualitative findings show that participants liked the program and benefited from it, but it seems that it worked to strengthen or consolidate the existing resources available to them. This is good. But can the same be said of population who may not have some of the resources available to these participants? For instance, one may have all the knowledge and the awareness about the importance of healthy diet for children. But if they are single parents and have limited or no income, it will be difficult to make that choice. In terms of the lack of the sustainability of the gains made with the intervention group at follow up, could the development of the children be a factor?
Finally, I suggest the authors do a thorough editing of the manuscript to correct some grammar issues. Below are some of the examples:
line 7 of the abstract, needs to read parenting skills regarding their
The last statement before the method “family habit” around seems incomplete.
In the text, where the authors state debate, discussion seem more appropriate to convey the intent and the process by which the intervention was implemented. Debate seems more like an argument about a given topic. But in the context of this manuscript, where participants talked and shared their experiences, discussion seems to be a better choice of word than debate.
Check the grammar on page 7, line 10. (abilities they worked during the sessions). Is this about skills or abilities they developed or built on?
Page 7, under 2.4.2 parenting style. Authoritative is repeated twice. I am assuming one of them is authoritarian.
Author Response

(The authors gave the same response as above.)

Round 2
Reviewer 1 Report
The authors elaborated most of comments. In my point of view the paper may be published in the proposed form.